# Ultrasound-Assisted Extraction of Taxifolin, Diosmin, and Quercetin from *Abies nephrolepis* (Trautv.) Maxim: Kinetic and Thermodynamic Characteristics

**DOI:** 10.3390/molecules25061401

**Published:** 2020-03-19

**Authors:** Mengxia Wei, Ru Zhao, Xiaojin Peng, Chunte Feng, Huiyan Gu, Lei Yang

**Affiliations:** 1College of Chemistry, Chemical Engineering and Resource Utilization, Northeast Forestry University, Harbin 150040, China; weimengxia@nefu.edu.cn (M.W.); zhaoru@nefu.edu.cn (R.Z.); pengxiaojin@nefu.edu.cn (X.P.); fctleyq@nefu.edu.cn (C.F.); 2Key Laboratory of Forest Plant Ecology, Ministry of Education, Northeast Forestry University, Harbin 150040, China; 3School of Forestry, Northeast Forestry University, Harbin 150040, China

**Keywords:** ultrasound-assisted extraction, kinetic, thermodynamic, *Abies nephrolepis*, flavonoids

## Abstract

Extraction behaviors of the 3 flavonoids taxifolin, diosmin, and quercetin have been investigated in *Abies nephrolepis* leaves and bark. The following operation parameters—ethanol volume fraction, liquid–solid ratio, temperature, ultrasound irradiation power and time, and ultrasound frequency—were varied to study their effect on the yield of the 3 flavonoids during extraction. The results showed that a low extraction efficiency occurred at 293.15 K due to slow kinetics, while the situation was significantly improved at 333.15 K. The kinetic data for the extraction yields of the 3 flavonoids achieved good fits by the first-order kinetic model. From the thermodynamic analysis results, we realized that the ultrasound-assisted extraction of taxifolin, diosmin, and quercetin from the leaves and bark of *A. nephrolepis* was a spontaneous and endothermic process in which the disorder increased (Δ*G*^0^ < 0, Δ*H*^0^ > 0, and Δ*S*^0^ > 0). According to the response surface methodology (RSM) analysis, under the optimal operation conditions (ethanol concentration of 50%, liquid–solid ratio of 20 mL/g, frequency of 45 kHz, extraction time of 39.25 min, ultrasound irradiation power of 160 W and temperature of 332.19 K), the total yield of the 3 flavonoids were 100.93 ± 4.01 mg/g from the leaves of *A. nephrolepis* (with 31.03 ± 1.51 mg/g, 0.31 ± 0.01 mg/g, 69.59 ± 2.57 mg/g for taxifolin, diosmin, and quercetin, respectively), and under the optimal operation conditions (ethanol concentration of 50%, liquid–solid ratio of 20 mL/g, frequency of 45 kHz, extraction time of 36.80 min, ultrasound irradiation power of 150 W and temperature of 328.78 K), 16.05 mg/g ± 0.38 mg/g were obtained from the bark of *A. nephrolepis* (with 1.44 ± 0.05 mg/g, 0.47 ± 0.01 mg/g, 14.14 ± 0.38 mg/g for taxifolin, diosmin, and quercetin, respectively), which were close to the prediction values.

## 1. Introduction

The evident pharmaceutical value of highly abundant plant medicines has confirmed that the leaves, flowers, stems, barks, and roots of plants in nature have good benefit for human beings and that these benefits mainly come from their bioactive compounds [1,2]. Extraction followed by purification was conducted for structural identification of the components and further significant bioactive applications. In general, optimized and valuable extraction processes have been obtained from numerous investigations aiming to research the influencing parameters and interaction among the affecting factors. Previous investigations of bioactive compound extraction processes usually aimed solely at one part of the plants such as leaves or bark, which suggested their applicability but often lacked widespread adoption. Different plant parts show variable cell structures, including the thickness and strength of the cell wall.

*Abies nephrolepis*, belonging to the family Pinaceae, is an important perennial timber arbor that is mainly distributed in the highlands of Northeast Asia, although distributions in Europe, North and Middle America and North Africa have also been mentioned [3]. *A. nephrolepis* trees have been widely planted in the Lesser Xingan Mountains (Heilongjiang, China) and Xiaowutai Mountain (Shanxi, China) regions, bringing remarkable economic benefits for those farmers [4,5]; additionally, the extra value of nonvolatile compounds and essential oils from this plant has received attention among academic circles. Previous studies have shown that nearly 30 flavonoids are found in *A. nephrolepis* leaves and bark, such as taxifolin, diosmin, and quercetin [6]. In addition, the taxifolin, diosmin, and quercetin showed broader commercial exploitation among the nearly 30 flavonoids, which gained widespread attention in medical community. Taxifolin (dihydroquercetin, Figure 1), 3,3′,4′,5,7-pentahydroxyflavanone, is also found in the xylem of *Larix gmelinii* [7], bark of *Cedrus deodara* [8], and seed of *Silybum marianum* [9]. Taxifolin has a noticeable positive impact on human health because it has cardioprotective [10], cytoprotective [11] and neuroprotective [12] activities, anti-inflammatory and antiallergic effects [13], clinically beneficial effects in the treatment of gastric injury [14] and so on. Diosmin, the diosmetin 7-*O*-rhamnosylglucoside (Figure 1), has been proposed as a drug due to its antitumor effects [15]. Furthermore, diosmin showed pharmacological antioxidant activity, including anti-oxidative stress, prevention of mitochondrial damage, and anti-myocardial infarction [16]. Quercetin, 3,3′,4′,5,7-pentahydroxyflavone (Figure 1), is mainly found in fruit and vegetables. It is one of the major flavonoids that is part of human diets. This compound has been confirmed to have beneficial anticancer effects [17]. Quercetin has also been used in the drug industry because of its inhibitory features in brain lesions [18], suppression of oxidative stress [19], inhibition of mesangial cell proliferation in early diabetic nephropathy [20], etc. Reports revealed that the phenolics from leaves of *A. nephrolepis* showed antioxidant in vitro [21] and anti-hepatoma in vivo [22].

In the past decade, ultrasonication has been of considerable interest as a newly developed reaction accelerator, and the growth of ultrasound-assisted extraction (UAE) as a novel extraction technology has quickly flourished owing its unique advantages [2,23]; numerous practical applications for bioactive compound extraction have been researched and successfully achieved, such as for flavonoids [24,25], anthocyanins [26], glycosides [27], coumarins [28], lignans [29], alkaloids [30], polysaccharides [31], and vegetable oils [32,33]. UAE is highly commended as a “green” extraction technology, owing to its characteristics of time savings, energy conservation, and environmental protection [34]. Furthermore, extremely expansive and grand prospect applications for UAE have been manifested in some special applications such as easily thermolabile and extreme low-temperature processes [35]. Nevertheless, previous extraction investigations mostly focused on the optimization of experimental operating conditions and their qualitative effects; the objective analyte mass and heat transfer process were less frequently reported [36]. We found no reports investigating the extraction process differences of taxifolin, diosmin, and quercetin with optimal operating parameters, especially regarding the mass transfer and heat transfer processes of these 3 flavonoids from the leaves and bark of *A. nephrolepis*.

In the present work, we systematically investigated the extraction behaviors of the 3 flavonoids taxifolin, diosmin, and quercetin from the leaves and bark of *A. nephrolepis*. The effects on the yields of the 3 flavonoids were studied by varying the following parameters: the ethanol volume fraction, liquid–solid ratio, reaction temperature, ultrasound irradiation power, ultrasound frequency, and ultrasound irradiation time. The mass transfer processes during the extraction of taxifolin, diosmin, and quercetin were analyzed by the equilibrium yield and rate constant under varying reaction temperatures, ultrasound irradiation powers and frequencies, and the heat transfer processes were analyzed under varying temperatures based on the thermodynamic parameters.

## 2. Results and Discussion

### 2.1. Effect of Single Factor

#### 2.1.1. Ethanol Volume Fraction

The hydroalcoholic proportion has a great impact on the cost consumption and extraction efficiency, and it should be investigated systematically. A series of ethanol concentrations (0−90%) were studied with the other operation parameters held constant: a liquid–solid ratio of 10 mL/g, temperature of 293.15 K, and ultrasound irradiation power, time, and frequency of 200 W, 30 min, and 45 kHz, respectively. The experimental yields of the 3 flavonoids from the leaves and bark are shown in Figure 2a,b. High ethanol concentrations promoted dissolution of the 3 flavonoids; in particular, an increasing trend was observed initially and slightly afterwards up to the 50% ethanol demarcation point, and the yield trends of flavonoids from both leaves and bark were similar. Consequently, 50% ethanol stood out as the best extraction solvent for the 3 flavonoids from both leaves and bark.

#### 2.1.2. Liquid–Solid Ratio

A small solvent volume is associated with inadequate extraction of the solid matrix; nevertheless, a large volume also introduces a large burden on the processing system and massive solvent consumption. The liquid–solid ratio was a significant factor that should be considered in improving the extraction yield of taxifolin, diosmin, and quercetin. A series of liquid–solid ratios (10–30 mL/g with intervals of 5 mL/g) were evaluated with the other operating conditions held constant (ethanol volume fraction of 50%, temperature of 293.15 K, and ultrasound frequency, power, and time of 45 kHz 200 W, 30 min, respectively). The experimental results regarding the effect of the liquid–solid ratio on the yields of these 3 flavonoids from the leaves and bark of *A. nephrolepis* are shown in Figure 3a,b, respectively. As shown in Figure 3a, the extraction efficiencies of the 3 flavonoids obviously increased in the range from 10 to 20 mL/g; however, with the addition of 50% ethanol, the extraction efficiency growth rate of the 3 flavonoids was subsequently slowed. As shown in Figure 3b, for quercetin, the yields increased obviously in the range from 10−25 mL/g, and with further addition of 50% ethanol, no obvious improvement of the yield was observed; for taxifolin and diosmin, the yields increased moderately with increasing liquid–solid ratio, and the increasing trend of the extraction efficiency of the 3 flavonoids was maintained at a low level. Considering the yields of the 3 flavonoids, 25 mL/g was the best liquid–solid ratio and was selected for the extraction of taxifolin, diosmin, and quercetin from the leaves and bark of *A. nephrolepis*. Comparing Figure 3a with Figure 3b, we deduced that the leaf tissue cytoderm was more easily permeated by solvent, leading to simply dissolution of the target analytes and a rapid mass transfer rate in contrast with that of the bark cytoderm; the tissue density may also be one reason for this result.

### 2.2. Analysis of Extraction Kinetics

#### 2.2.1. Under Various Ultrasound Irradiation Powers

The extraction kinetic model was applied to describe the target analyte yield variation with increasing time, to evaluate the experimental operating conditions affecting the target analyte yields and to predict the extraction rate of the overall process. The kinetic parameters were determined by assessing the mass transfer procedure of the target analyte molecules during the extraction process, and error analysis described the consistency between the kinetic model and experimental data, which demonstrated that extraction kinetic analysis was crucial during investigations. In the present work, four ultrasound irradiation powers (80−200 W with an interval of 40 W) were investigated, with the other operation conditions held constant (50% alcoholic solution-biomass ratio of 20 mL/g, ultrasound irradiation frequency of 45 kHz, and temperature of 293.15 K), to analyze the mass transfer difference of the taxifolin, diosmin, and quercetin molecules from the leaves and bark of *A. nephrolepis*. The yield of target analytes was measured by HPLC, with 1 mL aliquots analyzed immediately after being collected at given time intervals (0, 10, 20, 30, 40, 60, and 90 min) during the extraction process. The flavonoid yield experimental data fitted by first-order extraction kinetics are shown in Figure 4.

From this figure, first, we can see that the extraction yields of 3 types of flavonoids were well fitted by the first-order extraction kinetic model under varying experimental ultrasound irradiation power, and high dissolution production yields were obtained at 200 W. The results of the experimental data analysis implied that the extraction yields of the 3 flavonoids were well fitted with the first-order extraction kinetic model according to the coefficient of association *R*^2^ > 0.99. The results are well consistent with first-order extraction kinetics, which means the following: the extraction process of the 3 flavonoids can be explained by first-order extraction kinetics, the highest mass transfer resistance of the target analyte during the hydroalcoholic solvent extraction process is due to the rigid structure of the plant cell wall, and the extraction process can be seen as an irreversible consecutive process [37]. It should be mentioned that no interactions remained among the three flavonoid molecules after the dissolution process [38]. Second, over 90% of the flavonoid content could be obtained within 30 min for the leaves and bark; however, the extraction equilibrium yields of the 3 flavonoids were quite different under the same operation conditions. These results may be attributed to the tissue structure and cell wall permeability differences and/or content differences remaining in the leaves and bark. Third, for the dissolution process of individual flavonoids, quercetin, the most soluble component of the flavonoids investigated, had the highest average dissolution rate and maximum dissolution efficiency at 200 W, followed by taxifolin and diosmin. Ultrasound, as one kind of mechanical wave, causes cavitation, breaking up the plant cell cytoderm during the propagation process throughout the liquid medium, and the amplitude of ultrasound and ultrasound intensity are closely connected with the ultrasound irradiation power, which contributes to the varying yields of the target analytes under a variety of ultrasound irradiation powers [39]. Fourth, for the dissolution process of individual flavonoids, the molecules initially have a higher mass transfer efficiency, as expressed by the rapid increase in the yield, and finally, the production yield increases slowly and stabilizes. We also observed that the rate constant of the whole extraction process K (mg/g/min) became irregular with increasing ultrasound irradiation power. The concentration gradient between the liquid system and solid plant cell cultures and a limited amount of target analytes in plant biomass may be the primary reason for the variation in extraction rate and yield of objectives; the new solvent system generated a large density gradient between the solid and liquid, which increased with time, contributing to a decrease in the extraction rate and a stabilized yield of target analytes until reaching equilibrium [40].

#### 2.2.2. Under Various Ultrasound Frequencies

Many experimental operation factors may affect cavitation and sonication efficiency, such as ultrasound irradiation power and frequency, propagation liquid medium type and temperature [41]. The ultrasound frequency is an important experimental operation factor in the UAE process, and analysis of its influence on efficiency is a crucial procedure for sonication research. In this set of experiments, we evaluated variations in ultrasound frequencies (45 kHz, 80 kHz, and 100 kHz) on the effect of target flavonoid yields from the view of yield vs. time extraction kinetics. In the present investigation, certain experimental parameters were held constant: the ultrasound irradiation power was 200 W, the ultrasound propagation medium, ultrapure water, was maintained at a constant volume and temperature of 293.15 K, and the same batch of plastic round-bottom vessels were used, fixed in the same position to avoid differences in ultrasound intensity in the different experiments, in addition to the variety of ultrasound frequencies [25]. Figure 5 shows the experimental data fitted by the first-order extraction kinetic results.

From Figure 5, first, we can see that the flavonoid yields increased rapidly with increasing time from 0 to 10 min, then increased slightly as the time continued to 20 min and finally plateaued until 90 min, and the yield trend of flavonoids from both leaves and bark was similar. The extraction yields of the 3 flavonoids were well fitted by first-order extraction kinetics under varying experimental ultrasound frequencies, and a high product yield was obtained at 45 kHz. The results agree well with first-order extraction kinetics, which means that the extraction process of the 3 flavonoids could be illustrated by first-order extraction kinetics; the highest mass transfer resistance of the target analyte during the solvent extraction process is due to the rigid structure of the plant cell wall, and the extraction process can be seen as an irreversible consecutive process [37]. It should be mentioned that no interactions remained among the three flavonoid molecules after the dissolution process [38]. Additionally, over 90% of the flavonoid content could obtained within 30 min for the leaves and bark, which was consistent with the results for the variation in ultrasound irradiation power. Second, regarding the dissolution of individual flavonoids from the leaves of *A. nephrolepis* during extraction, the average dissolution rate changed slightly as the frequency was varied, and the maximum dissolution efficiency was at 45 kHz; at this frequency, the average dissolution rate and maximum dissolution equilibrium yield of the 3 flavonoids reached 0.19 mg/g/min, 0.0024 mg/g/min, and 0.14 mg/g/min for taxifolin, diosmin, and quercetin for both leaves and bark, respectively. In addition, a frequency of 45 kHz promoted better dissolution of the target flavonoid molecules than did the other two frequencies, with maximum equilibrium yields of 18.49 mg/g, 0.16 mg/g, and 40.56 mg/g for taxifolin, diosmin, and quercetin from the leaves, respectively. Third, regarding the dissolution of the individual flavonoids, the molecules initially have a higher mass transfer rate, as expressed by the rapid increase in the production yield, and then, the yield increases slowly and stabilizes; it should be mentioned that the rate constant of the whole extraction process, K (mg/g/min), was constant under variation ultrasound frequency despite the flavonoid extraction efficiency increase. Furthermore, the rate constant of the whole extraction process was not much different between leaves and bark for individual flavonoids, and the yield trends of the flavonoids from both leaves and bark were similar. Finally, higher flavonoid dissolution rates and maximum yields were more easily obtained from the leaves of *A. nephrolepis* than from the bark. For the dissolution of quercetin from the bark of *A. nephrolepis*, a 0.14 mg/g/min rate constant of the whole extraction process was obtained, which was equal to the rate constant of the whole extraction process achieved from the leaves of *A. nephrolepis*. However, the quercetin equilibrium yield from bark was 7.87 mg/g at 45 kHz, which was much less than the 40.56 mg/g equilibrium yield obtained from the leaves of *A. nephrolepis*, and we deduced that the leaf tissue cytoderm was more easily permeated by solvent, leading to simply dissolution of the target analytes and a rapid mass transfer rate in contrast with that of the bark cytoderm; the tissue density may also be one reason for this result. Finally, we conclude that a high flavonoid yield was more easily obtained at 45 kHz, and the extraction kinetic rate constant of the 3 flavonoids remained constant under various ultrasound frequencies. The production yields of the target analytes differ under the various ultrasound frequencies, which may be because the formation of cavitation bubbles is difficult within the given time at higher ultrasound frequencies [42]. Another reason may be that the frequency of 45 kHz more efficiently avoided the production of radicals within the cavitation bubbles from the sonication procedure [43]. The yield of target analytes differs in a variety of tissues, which may be explained by the variation in tissue density between leaves and bark.

#### 2.2.3. Under Various Reaction Temperatures

Temperature is another key parameter in molecule propagation during ultrasound incubation and plays an important role in the UAE process. In the present research investigation, a series of temperatures (293.15−333.15 K) were used to determine the influence on the target flavonoid extraction kinetics. In this batch experiment, other experimental conditions were held constant except the temperature of the ultrasound propagation medium, with an ultrasound power, frequency, and time of 200 W, 45 kHz, and 90 min, respectively. In addition, in this set of experiments, assays were conducted in the same batch of plastic round-bottom vessels and a constant volume of the ultrasound propagation medium, ultrapure water, was maintained in the ultrasound bath. Because cavitation causes an increase in temperature, it is indispensable to adopt measures to avoid it. The required stationary temperature of the ultrasound propagation medium was achieved by a constant rate of injection and discharge of ultrapure water. The experimental data of the flavonoid yield from the leaves and bark of *A. nephrolepis* fitted by first-order extraction kinetics are displayed in Figure 6a,b, respectively.

As shown in Figure 6, first, we can see that the yield of taxifolin, diosmin, and quercetin from the leaves and bark of *A. nephrolepis* varied with the ultrasound irradiation time extension and reaction temperature. The extraction yield of the 3 flavonoids increased rapidly over time in the former 10 min and then much more slowly in the next 10 min, and with increasing ultrasound irradiation time, the yield of the 3 flavonoids did not obviously improve, reaching a limit at 90 min for a given reaction temperature; and the yield trend of flavonoids from both leaves and bark was similar. Moreover, the extraction yield of the 3 flavonoids were well fitted by first-order extraction kinetics under various experimental reaction temperatures according to the correlation coefficient *R*^2^ > 0.99, and a high product yield was obtained at 333.15 K. Second, for the dissolution of individual flavonoids from the leaves of *A. nephrolepis*, the average dissolution rate changed under various extraction temperatures, and the maximum dissolution efficiency was at 333.15 K. The reaction temperature of 333.15 K promoted better dissolution of the target flavonoid molecules than did the other four reaction temperatures, with maximum equilibrium yields of 30.46 mg/g, 0.27 mg/g, and 66.59 mg/g for taxifolin, diosmin, and quercetin from the leaves of *A. nephrolepis*, respectively, and 1.37 mg/g, 0.43 mg/g, and 13.15 mg/g from the bark. Finally, higher flavonoid maximum yields were more easily obtained from the leaves of *A. nephrolepis* than from the bark, and the dissolution rate constants were irregular under various experimental temperatures. For quercetin dissolution from the leaves of *A. nephrolepis*, a 66.59 mg/g maximum quercetin equilibrium yield could be obtained at 333.15 K; however, only 13.15 mg/g could be obtained under the same operating conditions from the bark. Regarding quercetin dissolution from the leaves, for the rate constant K of the whole extraction process, values of 0.14 mg/g/min, 0.13 mg/g/min, 0.13 mg/g/min, 0.11 mg/g/min, and 0.12 mg/g/min were obtained from 293.15 K to 333.15 K, which means that the extraction rate constant of the whole extraction process was irregular with increasing temperature. Under the operating conditions, the maximum extraction yields of taxifolin, diosmin, and quercetin were 30.97 ± 1.55 mg/g, 0.27 ± 0.01 mg/g, and 67.47 ± 3.37 mg/g from the leaves of *A. nephrolepis*, respectively, and 1.40 ± 0.07 mg/g, 0.44 ± 0.00 mg/g, and 13.27 ± 0.63 mg/g from the bark. Finally, we conclude that compared with other experimental temperatures, 333.15 K could readily promote a high flavonoid yield, and the extraction kinetic rate constant of the 3 flavonoids was irregular under varying temperatures. The target analyte yields differed in a variety of tissues, which may be explained by the variation in tissue density between leaves and bark. The predicted equilibrium yield and average extraction rate constant of the objective analytes varied under various temperatures because the thermal energy increased the diffusion and dissolution of the target compounds and other constituent molecules; in addition, higher temperatures affect the pressure, viscosity, and surface tension of the liquid system, and these properties may benefit acoustic wave propagation [44].

### 2.3. The Thermodynamic Parameters

For careful evaluation and development of the flavonoids extracted from leaves and bark of *A. nephrolepis*, analysis of the thermodynamic parameters Δ*G*^0^ (kJ/mol), Δ*H*^0^ (kJ/mol) and Δ*S*^0^ (J/mol/K) is crucial to fully comprehend the heat transfer process during extraction of the 3 flavonoids. In this set of experiments, temperatures ranging from 293.15 K to 333.15 K were investigated to obtain experimental data of the three target flavonoids and calculate the three thermodynamic parameters. In the present work, certain parameters were kept constant, such as an ultrasound irradiation power of 200 W and a frequency of 45 kHz, under varied experimental temperatures, and the same batch of experiments was conducted in the same batch of round-bottom vessels, which were maintained at the same position during the whole extraction process, and a constant volume of ultrapure water was maintained in the ultrasound bath. It is essential to take actions to keep the temperature constant because cavitation increases the temperature. A stationary temperature in the ultrasound bath was achieved by a constant rate of injection and discharge of ultrapure water. From Equation (5), we can obtain the linear parameter information about Δ*H*^0^ and Δ*S*^0^, and Δ*G*^0^ can be obtained according to Equation (7). The experimental data and calculation results for the thermodynamic parameters and error analysis are shown in Table 1. From Table 1, first, we can see that for the experimental and calculated data for the extraction of 3 flavonoids from the leaves and bark of *A. nephrolepis* under varied operating temperatures, the correlation coefficient *R*^2^ was above 0.98, which indicated a good linear relationship between *Lnk* and 1/*T* (Equation (5)). Second, the equilibrium coefficient *k* was higher at 333.15 K than at 293.15 K, which was consistent with the yield of flavonoids increasing with increasing temperature. Third, the positive value of Δ*H*^0^ and Δ*S*^0^ means that the extraction of the 3 flavonoids is an endothermic process in which the disorder increases. Furthermore, the extraction process of the 3 flavonoids is feasible because of the negative value of Δ*G*^0^. Additionally, the Δ*G*^0^ values were −20.94 kJ/mol, −21.65 kJ/mol, −22.36 kJ/mol, −23.08 kJ/mol, and −23.79 kJ/mol, for taxifolin at 293.15 K, 303.15 K, 313.15 K, 323.15 K, and 333.15 K, respectively. The absolute value of the Gibbs free energy increased, proving that the flavonoid extraction process is thermodynamically spontaneous [45]. Similar results were obtained for the other two flavonoids extracted from the leaves of *A. nephrolepis* and other target objectives extracted from the bark of *A. nephrolepis*. Thermodynamically, high temperature is more likely to promote endothermic processes, mass transfer is easily facilitated, and diffusion and dissolution of the target objective molecules occur more easily [46,47]. Finally, we conclude that the extraction of the target analytes from both the leaves and bark of *A. nephrolepis* is a spontaneous and endothermic process in which the disorder increases.

### 2.4. Optimization of Extraction Procedure

For the 3 flavonoids extraction procedure analysis from leaves of *A. nephrolepis*, from the analysis of variance (ANOVA) results (showed in Table 2) we observed that the model is significant with *F*-value of 16.15, only 0.07% chance that a “Model *F*-Value” this large could occur large because of noise. The *p* value of *X*_2_, *X*_3_, *X*_1_^2^, *X*_2_^2^ and *X*_3_^2^ were less than 0.05 indicated that the three model terms were significant, and on the contrary other 4 model terms were non-significant. The “Lake of fit” of was non-significant with the *p* value of 0.660 implied the Lack of Fit was non-significant and the developed models could be applied to predict the responses. High correlation could be observed between the experimental data and the predicted value with “Adj *R*^2^” of 0.8950. The “Adeq Precision” value of 11.019 greater than 4 could conclude that an adequate signal was obtained [48]. For the 3 flavonoids extraction procedure analysis from bark of *A. nephrolepis*, from the ANOVA results we observed that the model is significant with *F*-value of 10.95, only 0.23% chance that a “Model *F*-Value” this large could occur large because of noise. The *p* value of *X*_1_, *X*_3_, *X*_1_^2^, and *X*_3_^2^ were less than 0.05 indicated that the model terms were significant, and on the contrary other 5 model terms were non-significant. The “Lake of fit” of was non-significant with the *F*-value of 10.95, and implied the Lack of Fit was non-significant and the developed models could be applied to predict the responses. High correlation could be observed between the experimental data and the predicted value with “Adj *R*^2^” of 0.8484. The “Adeq Precision” value of 8.575 greater than 4 could conclude that an adequate signal was obtained. Finally, the fitting quadratic equation fitted by the experiments data was showed in Equation (1) for leaves and Equation (2) for bark.
(1)$$Y_{\mathrm{Leaves}}=100-1.70X_{1}-3.25X_{2}+9.13X_{3}+2.61X_{1}X_{2}+0.98X_{1}X_{3}+3.13X_{2}X_{3}-6.78{X_{1}}^{2}-4.05{X_{2}}^{2}-4.87{X_{3}}^{2}$$
(2)$$Y_{\mathrm{Bark}}=16.89-1.40X_{1}-0.039X_{2}+2.16X_{3}+0.92X_{1}X_{2}-1.16X_{1}X_{3}-0.89X_{2}X_{3}-3.61{X_{1}}^{2}-1.49{X_{2}}^{2}-2.48{X_{3}}^{2}$$

#### 2.4.1. Analysis of the Response Contour

The interaction between two variables of the extraction time (*X*_1_), ultrasound irradiation power (*X*_2_) and temperature (*X*_3_) with the remaining parameter fixed at 0 level on the influence of the extraction total yield of the 3 flavonoids from leaves and bark were showed in Figure 7. As we can see from Figure 7(a1), with the *X*_1_ and *X*_2_ continuously increasing, the *Y_Leaves_* trend obtained with the increasing initially and then decreasing finally, similar trend was obtained in Figure 7(b1). Figure 7(a2) indicated the interaction effect on *X*_1_ and *X*_3_ fixed *X*_2_ zero level on the extraction total yield of 3 flavonoids *Y_Leaves_* from the leaves of *A. nephrolepis*. *X*_3_ is significant factors which have significant influences on *Y_Leaves_*, and continuously increasing trend obtained with the increasing of *X*_3_, *X*_1_ have comparatively less influences on *Y_Leaves_*, and similar trend were obtained in Figure 7(b2, a3, and b3).

#### 2.4.2. Verification Tests

For the procedure of 3 flavonoids extracted from the leaves of *A. nephrolepis*, the optimal operation conditions (extraction time of 39.25 min, ultrasound irradiation power of 156.91 W and temperature of 332.19 W) was estimated, and the total yield of the 3 flavonoids was estimated 104.32 mg/g. Verification test was conducted under the estimation condition (ethanol concentration of 50%, liquid–solid ratio of 20 mL/g, frequency of 45 kHz, extraction time of 39.25 min, ultrasound irradiation power of 160 W and temperature of 332.19 K), and the total yield of the 3 flavonoids were 100.93 mg/g ± 4.01 mg/g for leaves of *A. nephrolepis* (with 31.03 ± 1.51 mg/g, 0.31 ± 0.01 mg/g, 69.59 ± 2.57 mg/g for taxifolin, diosmin, and quercetin, respectively), which were close to the prediction values.

For the procedure of 3 flavonoids extracted from the bark of *A. nephrolepis*, the optimal operation conditions (extraction time of 36.80 min, ultrasound irradiation power of 148.74 W and temperature of 328.78 K) was estimated, and the total yield of the 3 flavonoids was estimated 17.73 mg/g. Verification test was conducted under the estimation condition (ethanol concentration of 50%, liquid–solid ratio of 20 mL/g, frequency of 45 kHz, extraction time of 36.80 min, ultrasound irradiation power of 150 W and temperature of 328.78 K), and the total yield of the 3 flavonoids were 16.05 mg/g ± 0.38 mg/g for bark of *A. nephrolepis* (with 1.44 ± 0.05 mg/g, 0.47 ± 0.01 mg/g, 14.14 ± 0.38 mg/g for taxifolin, diosmin, and quercetin, respectively), which were close to the prediction values.

## 3. Materials and Methods

### 3.1. Chemicals and Materials

The reference substances taxifolin and quercetin, with purity above 98%, were obtained from Shanghai Yuanye Bio-Technology Co., Ltd. (Shanghai, China), and diosmin, with purity above 99%, was purchased from Ziyi Chemical Reagent Factory (Shanghai, China); chromatographic grade acetonitrile and phosphoric acid were obtained from Thermo Fisher Scientific (Shanghai, China). A 0.45 µm filtration was needed for the standards, flavonoid extract samples and eluent, and some samples were diluted to appropriate concentrations by pure water before HPLC analysis.

Leaves and bark of *A. nephrolepis* were provided by the Liangshui Experimental Forest Farm (Northeast Forestry University, Yichun, Heilongjiang, China) and collected from one tree of ten years of growth in September 2017. The leaves and bark of *A. nephrolepis* specimens were certified by Prof Huiyan Gu (Northeast Forestry University, Harbin, China). The materials were dried at ambient temperature for 2 weeks, ground into a homogeneous powder, and then sieved (60–80 mesh). The sieved samples were stored in polyethylene zipper bags at 4 °C until use. The same batch of sample was used in all experiments.

### 3.2. Experimental Apparatus

An ultrasonic bath ((KQ-200VDB, Kunshan, China) is the main processing apparatus for accelerating the extraction, which could be set to a certain ultrasound irradiation power (from 80 to 200 W with intervals of 40 W) and three given frequencies of 45, 80, and 100 kHz; the apparatus was equipped with a poor temperature control system, which was improved by a pump to displace the inlet and outlet water.

### 3.3. Ultrasound-Assisted Extraction Procedure

Dried powder of *A. nephrolepis* leaves or bark was mixed with a volume of ethanol solution in the same batch of plastic round-bottom vessels. Ultrasound treatment was followed by filtration after cooling and then HPLC analysis. Varying ethanol concentrations (0−90% (*v*/*v*)) and liquid–solid ratios (10−30 mL/g) were analyzed initially, and the extraction kinetics of the 3 flavonoids were evaluated systematically under various ultrasound powers (80−200 W), three frequencies (45 kHz, 80 kHz, 100 kHz) and various temperatures (293.15−333.15 K). Finally, the thermodynamic analysis was carried out in a temperature range from 293.15 K to 333.15 K.

### 3.4. Modeling of Extraction Kinetics

To analyze and predict the extraction process on a large scale, it is indispensable to discuss the relevant extraction kinetic model. Based on mass transfer theory, during extraction of the target analytes, the highest mass transfer resistance of the process involves the rigid structure of the plant cell wall, and the extraction can be considered to be an irreversible consecutive process [37]. Furthermore, no interactions were observed between the objectives, and it can be concluded that the extraction process can be fitted by first-order extraction kinetics [38]; the general equation model can be written as:*Ln*(*Y_e_* − *Y_t_*) = *LnY_e_* − *K_t_*(3)
*Y_t_* = *Y_e_*[1 − *exp*(−*K_t_*)](4)
where *Y_t_* and *Y_e_* are the taxifolin, diosmin, and quercetin yield at any time (min) and at equilibrium (mg/g), respectively, and *K* (mg/g/min) is the rate constant of the whole extraction process.

### 3.5. Thermodynamic Characteristics

The taxifolin, diosmin, and quercetin extraction data were fitted by thermodynamic analysis under varying temperatures. The thermodynamic van’t Hoff equation for describing the extraction process under varying temperature is shown as [49]:*Lnk* = −Δ*H*^0^/*RT* + Δ*S*^0^/*R*(5)
*k* = *Y_e_*/(*Y_total_* − *Y_e_*)(6)
Δ*G*^0^ = Δ*H*^0^ − Δ*S*^0^(7)

*k* is the distribution coefficient for a liquid–solid system, which is calculated using Equation (6); *Y_e_* and *Y_total_* are the target individual objective yield at equilibrium (mg/g) and the total extracted and unextracted yield of target analytes, respectively, and the total yield of target analytes was calculated by the combination of the extracts three times experiments with a 50% ethanol volume fraction at 293.15 K, 200 W and 45 kHz, in which every extraction procedure lasted 90 min to achieve effective extraction; and Δ*G*^0^, Δ*H*^0^ and Δ*S*^0^ are the thermodynamic parameters representing the Gibbs free energy change (kJ/mol), enthalpy change (kJ/mol) and entropy change (J/mol/K), respectively.

Positive and/or negative values of the Δ*H*^0^ enthalpy change represent the endothermic and/or exothermic characteristics, and the Δ*S*^0^ entropy change reveals that the flavonoid molecule extraction process is reversible and/or irreversible. These two parameters were obtained by the linear relationship using *Lnk* vs. 1/*T*, which were calculated by the slope value and vertical intercept of the straight line. In addition, Δ*G*^0^ obtained from Equation (7) was used to evaluate whether the process was spontaneous or nonspontaneous.

### 3.6. HPLC Instruments and Quantitative Conditions

Qualitative and quantitative analyses were performed by HPLC with an Agilent 1260 system (USA). Chromatographic separation was carried out on an Agilent Extend-C18 reversed-phase column (4.6 mm × 250 mm, 5 μm), and the elution system for chromatographic separation consisted of a gradient of 5‰ phosphoric acid in ultrapure water (A pump) and acetonitrile (B pump) according to the following conditions: 0–40 min, 15–30% B. The injection volume was 10 μL with an eluant flow rate of 1 mL/min and column temperature of 25 °C. Evaluation of the retention times by running the samples after the addition of pure reference standards helped to analyze the target analytes. A UV–vis spectrophotometer set to a UV wavelength of 294 nm was used for detecting taxifolin, a wavelength of 225 nm was used for diosmin and a wavelength of 375 nm was used for quercetin. The run time was set to 40 min. Under the above conditions, the retention times of the absorption peaks were 14.6 min for taxifolin, 16.6 min for diosmin, and 32.5 min for quercetin. In addition, the chromatography analysis of mixture standard, *A. nephrolepis* leaves and bark were showed in Figure 8.

Stock solutions of 2.00 mg/mL taxifolin, diosmin, and quercetin dissolved in methanol were placed in a refrigerator and stored until use. Subsequently, the samples were prepared at certain concentrations, as necessary. According to the HPLC analysis results, we obtained calibration curves for the respective peak areas of taxifolin, diosmin, or quercetin (*Y*) and their concentrations (*X*):*Y_taxifolin_* = 469.8*X* + 2.9 (*R*^2^ = 0.9923, *n* = 7),(8)
*Y_diosmin_* = 4690.6*X* + 3.6 (*R*^2^ = 0.9992, *n* = 7) and(9)
*Y_quercetin_* = 80,470.0*X* + 424.5 (*R*^2^ = 0.9999, *n* = 7).(10)

Good linear correlation was achieved for taxifolin (0.007–0.4 mg/mL), diosmin (0.0008–0.08 mg/mL), and quercetin (0.017–1.7 mg/mL).

### 3.7. Optimization of Extraction Procedure with RSM

A Box–Behnken design of RSM was employed to predict the optimal total flavonoids extraction conditions in regard to three factors, which include extraction time (*X*_1_; 30–50 min), ultrasound irradiation power (*X*_2_; 120–200 W) and temperature (*X*_3_; 313.15–333.15 K), with other parameters fixed (liquid–solid ratio of 20 mL/g; ethanol concentration of 50%; ultrasound frequency of 45 kHz). The factorial design is composed of 17 runs (12 factorial runs and 5 center runs). Total flavonoids yield was set for the response to combine the independent factors (Appendix A). Experiments were performed at random to minimize the impacts of unforeseen variability in the determined responses. Each factorial run was conducted in triplicate and the average values were presented as actual values. The interactions between the responses and the three independent variables were evaluated by the generalized form of a quadratic equation as Equation (11) [50,51]:(11)$$Y=\beta_{0}+\sum_{i=1}^{3}\beta_{i}X_{i}+\sum_{i=1}^{3}\beta_{ii}X_{i}{}^{2}+\sum_{i=1}^{2}\sum_{j=i+1}^{3}\beta_{ij}X_{i}X_{j}$$
where *Y* is the predicted response; *β*_0_, *β**_i_*, *β_ii_* and *β_ij_* are regression coefficients of the intercept, linear, quadratic, and interaction terms, respectively; and *X_i_* and *X_j_* are coded independent parameters.

## 4. Conclusions

In the present study, the kinetics of 3 flavonoids, taxifolin, diosmin, and quercetin, from the leaves and bark of *A. nephrolepis* were accomplished at different ultrasound irradiation powers (80, 120, 160, and 200 W), ultrasound frequencies (45, 80, and 100 kHz), and temperatures (293.15, 303.15, 313.15, 323.15, and 333.15 K) using an ethanol aqueous solution (50% volume fraction and 20 mL/g liquid–solid ratio). The experimental results revealed that the leaves of *A. nephrolepis* contained more taxifolin, diosmin, and quercetin than did the bark. The experimental data were well fitted with first-order extraction kinetics, which indicated that the extraction process could be explained by the first-order extraction rate constant and the predicted equilibrium yield of the objective analytes. High yields of the 3 target flavonoids were promoted by high ultrasound power (200 W), high temperature (333.15 K) and low ultrasound frequency (45 kHz), which led to high equilibrium yield for taxifolin, diosmin, and quercetin among the various experimental operation conditions. From the thermodynamic analysis results, we realized that the UAE of taxifolin, diosmin, and quercetin from the leaves and bark of *A. nephrolepis* was a spontaneous and endothermic process in which the disorder increased (Δ*G*^0^ < 0, Δ*H*^0^ > 0, and Δ*S*^0^ > 0). Additionally, the Δ*G*^0^ absolute value increased with increasing ultrasound propagation medium temperature, meaning that the extraction process of the 3 flavonoids was more spontaneous and feasible when using a high temperature (333.15 K). This experiment can be useful for the extraction behavior analysis of flavonoids and other bioactive compounds and their mass transfer and heat transfer from plant tissues.

## Figures and Tables

**Figure 1 molecules-25-01401-f001:**
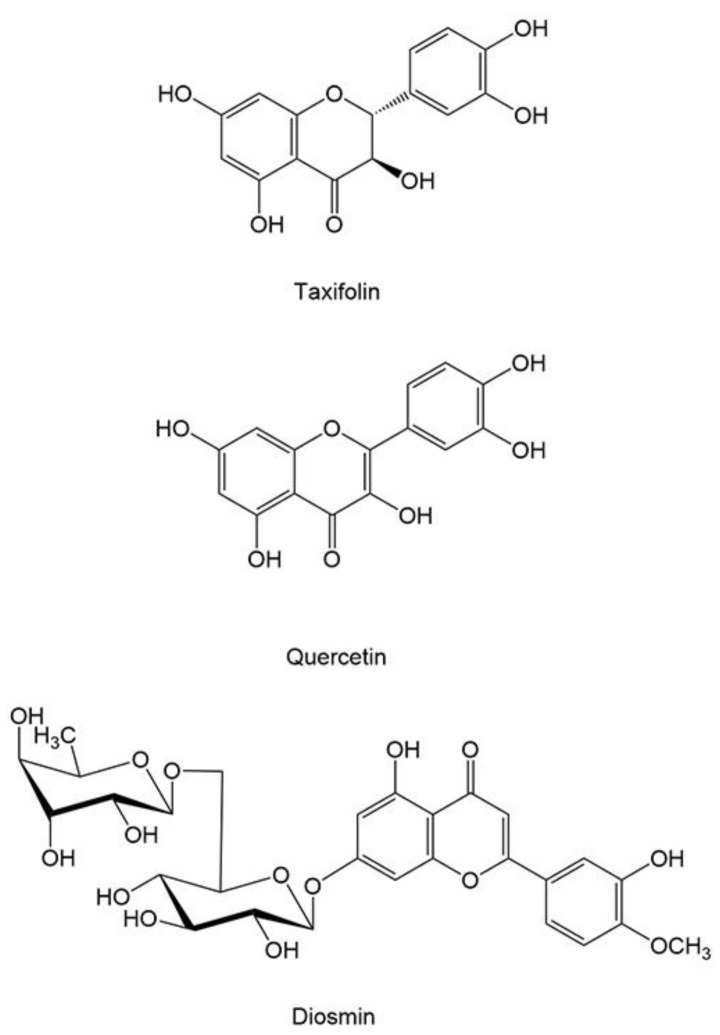
The molecular structure of taxifolin, quercetin, and diosmin.

**Figure 2 molecules-25-01401-f002:**
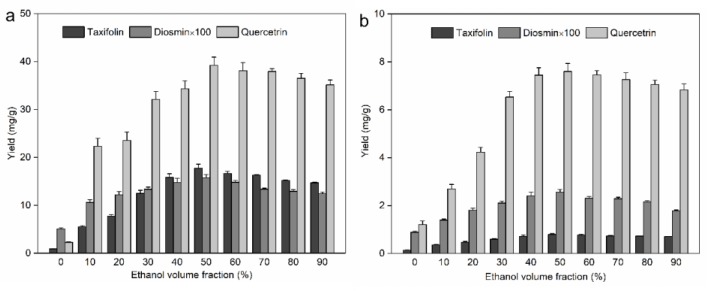
Effect of ethanol volume fraction on the yield of taxifolin, diosmin, and quercetin from *Abies nephrolepis* leaves (**a**) and bark (**b**).

**Figure 3 molecules-25-01401-f003:**
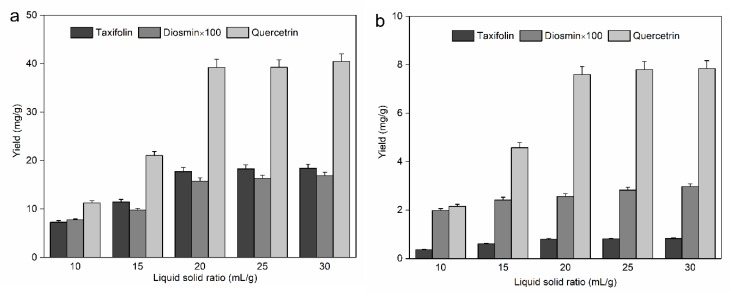
Effect of liquid–solid ratio on the yield of taxifolin, diosmin, and quercetin from *Abies nephrolepis* leaves (**a**) and bark (**b**).

**Figure 4 molecules-25-01401-f004:**
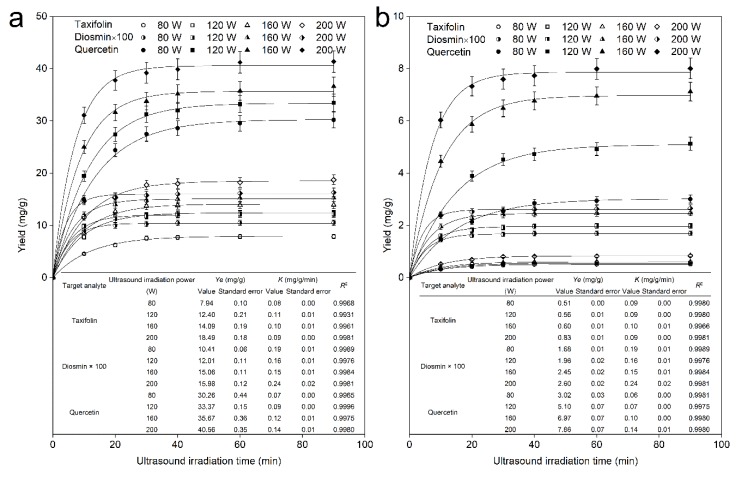
Dynamic curves for taxifolin, diosmin, and quercetin with various ultrasound irradiation powers (80 W, 120 W, 160 W, and 200 W) from *Abies nephrolepis* leaves (**a**) and bark (**b**).

**Figure 5 molecules-25-01401-f005:**
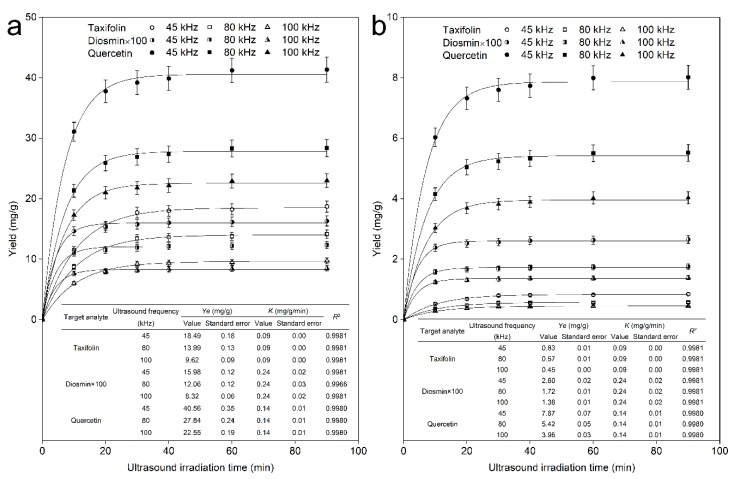
Dynamic curves for taxifolin, diosmin, and quercetin with various ultrasound frequencies (45 kHz, 80 kHz, and 100 kHz) from *Abies nephrolepis* leaves (**a**) and bark (**b**).

**Figure 6 molecules-25-01401-f006:**
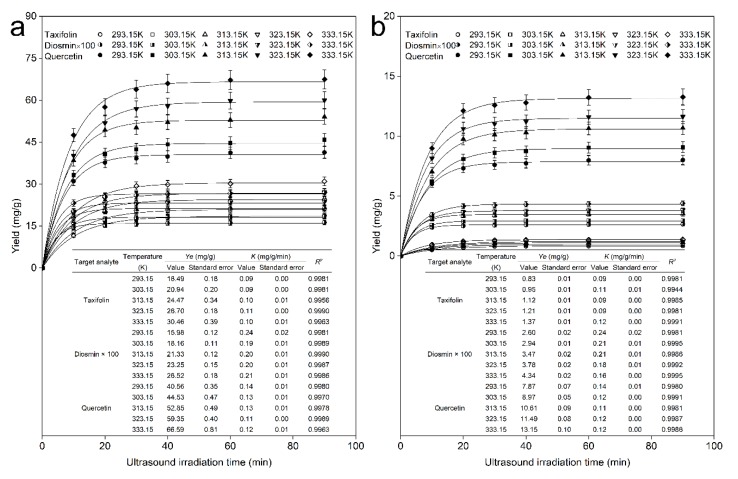
Dynamic curves for taxifolin, diosmin, and quercetin with various reaction temperatures (293.15 K, 303.15 K, 313.15 K, 323.15 K, 333.15 K, and 343.15 K) from *Abies nephrolepis* leaves (**a**) and bark (**b**).

**Figure 7 molecules-25-01401-f007:**
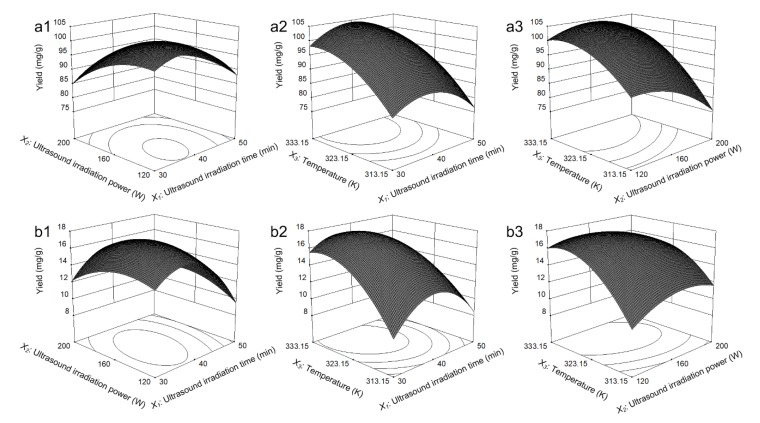
Response surface and contour plots. (**a1**) Effect of extraction time and ultrasound irradiation power on the extraction total yields of the 3 flavonoids from the leaves of *A. nephrolepis*, (**a2**) effect of extraction time and temperature on the extraction total yields of the flavonoids from the leaves of *A. nephrolepis*, (**a3**) Effect of ultrasound irradiation power and temperature on the extraction total yields of the 3 flavonoids from the leaves of *A. nephrolepis*, (**b1**) Effect of extraction time and ultrasound irradiation power on the extraction total yields of the 3 flavonoids from the bark of *A. nephrolepis*, (**b2**) effect of extraction time and temperature on the extraction total yields of the flavonoids from the bark of *A. nephrolepis*, (**b3**) Effect of ultrasound irradiation power and temperature on the extraction total yields of the 3 flavonoids from the bark of *A. nephrolepis*.

**Figure 8 molecules-25-01401-f008:**
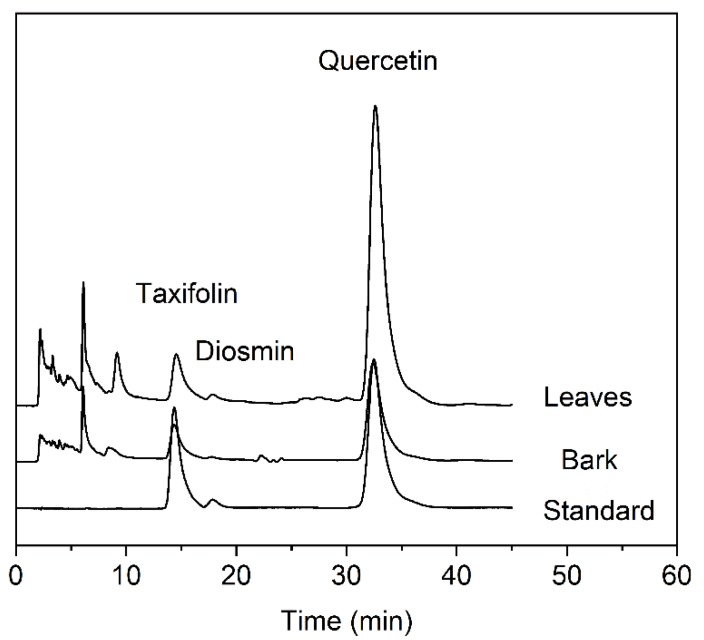
The chromatography analysis of mixture standard, *A. nephrolepis* leaves and bark (375 nm).

**Table 1 molecules-25-01401-t001:** Thermodynamic parameters of target flavonoids extracted from leaves and barks of *A. nephrolepis*.

Plant Part	Target Analyte	Temperature (K)	Parameters				
			*k* (mg/g/min)	*R* ^2^	∆*G* (kJ/mol)	∆*H* (kJ/mol)	∆*S* (J/mol K)
Leave	Taxifolin	293.15	0.69	0.9871	−20.94	21.92	71.41
		303.15	0.87		−21.65		
		313.15	1.19		−22.36		
		323.15	1.45		−23.08		
		333.15	2.08		−23.79		
	Diosmin	293.15	0.74	0.9861	−22.71	23.57	77.56
		303.15	0.94		−23.49		
		313.15	1.32		−24.26		
		323.15	1.63		−25.04		
		333.15	2.41		−25.81		
	Quercetin	293.15	0.83	0.9788	−25.16	25.85	85.90
		303.15	0.99		−26.01		
		313.15	1.44		−26.87		
		323.15	1.97		−27.73		
		333.15	2.91		−28.59		
Bark	Taxifolin	293.15	0.02	0.9943	−0.62	10.31	2.14
		303.15	0.02		−0.64		
		313.15	0.03		−0.66		
		323.15	0.03		−0.68		
		333.15	0.03		−0.70		
	Diosmin	293.15	0.57	0.9876	−18.49	19.95	63.14
		303.15	0.70		−19.12		
		313.15	0.94		−19.75		
		323.15	1.12		−20.38		
		333.15	1.55		−21.01		
	Quercetin	293.15	0.80	0.9827	−24.80	25.50	84.68
		303.15	1.02		−25.65		
		313.15	1.49		−26.49		
		323.15	1.84		−27.34		
		333.15	2.87		−28.19		

**Table 2 molecules-25-01401-t002:** Analysis of variance (ANOVA) for response surface quadratic model, and fit statistics for response values ^a^.

Source ^a^	Sum of Squares		Df		Mean Square		*F*		*p*			
	*Y_Leaves_* ^c^	*Y_Bark_*	*Y_Leaves_*	*Y_Bark_*	*Y_Leaves_*	*Y_Bark_*	*Y_Leaves_*	*Y_Bark_*	*Y_Leaves_*	*Y_Bark_*		
Model	1246.53	164.04	9	9	138.50	18.23	16.15	10.95	0.0007	0.0023		Significant ^d^
*X* _1_ ^b^	23.23	15.58	1	1	23.23	15.58	2.71	9.35	0.1438	0.0184		
*X* _2_	84.38	0.01	1	1	84.38	0.01	9.84	0.01	0.0165	0.9343		
*X* _3_	666.56	37.43	1	1	666.56	37.43	77.71	22.47	<0.0001	0.0021		
*X* _1_ *X* _2_	27.18	3.38	1	1	27.18	3.38	3.17	2.03	0.1183	0.1972		
*X* _1_ *X* _3_	3.84	5.38	1	1	3.84	5.38	0.45	3.23	0.5248	0.1152		
*X* _2_ *X* _3_	39.09	3.20	1	1	39.09	3.20	4.56	1.92	0.0702	0.2083		
*X* _1_ ^2^	193.40	54.95	1	1	193.40	54.95	22.55	33.00	0.0021	0.0007		
*X* _2_ ^2^	68.93	9.30	1	1	68.93	9.30	8.04	5.58	0.0252	0.0501		
*X* _3_ ^2^	100.05	25.83	1	1	100.05	25.83	11.66	15.51	0.0112	0.0056		
Residual	60.04	11.66	7	7	8.58	1.67						
Lack of Fit	18.11	0.23	3	3	6.04	0.08	0.58	0.03	0.6608	0.9933		Not significant
Pure Error	41.93	11.43	4	4	10.48	2.86						
Cor Total	1306.57	175.70	16	16								
	Std. dev.	Mean	C.V. %	PRESS	*R* ^2^	Adj *R*^2^	Pred *R*^2^	Adeq precision				
*Y_Leaves_*	2.93	92.61	3.16	355.28	0.9540	0.8950	0.7281	11.0190				
*Y_Bark_*	1.29	13.33	9.68	21.47	0.9337	0.8484	0.8778	8.5750				

^a^ The results were obtained with Design Expert 8.0.6 software. ^b^
*X*_1_ is the extraction time (min); *X*_2_ is the ultrasonic irradiation power (W); *X*_3_ is the temperature (K). ^c^
*Y_Leaves_* is the total yield of 3 flavonoids from leaves of *A. nephrolepis*; *Y_Bark_* is the total yield of 3 flavonoids from bark of *A. nephrolepis*. ^d^ Significant at *p* < 0.05.

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
