# Peer review of "Ultrasound-Assisted Extraction of Taxifolin, Diosmin, and Quercetin from Abies nephrolepis (Trautv.) Maxim: Kinetic and Thermodynamic Characteristics"

_molecules, 2020, doi:10.3390/molecules25061401_

Round 1
Reviewer 1 Report
The manuscript of Wei et al. reports data on extraction, carried out through ultrasound assisted maceration, of diosmin, quercetin and taxifolin. In particular, the detailed and well discussed influence of different factors and parameters was deeply investigated. Variation in ultrasound power, frequency and temperature could modulate the extraction kinetix of three bioactive substances. Thus, studying their behavior in the different ultrasound conditions could favorably augment substances' recovery and their uses.
The manuscript is well organized, and it has a good scientific soundness.
Please make Figures more clear
Author Response
Dear Reviewer,
Thank you for your comments concerning our manuscript entitled “Ultrasound-assisted extraction of taxifolin, diosmin and quercetin from Abies nephrolepis (Trautv.) Maxim.: kinetic and thermodynamic characteristics” (ID: molecules-745420). Those comments are all valuable and very helpful for revising and improving our paper, as well as the important guiding significance to our researches. We have studied comments carefully and have made correction which we hope meet with approval. All revisions made to the manuscript have been marked up with the "Track Changes" function in the revised manuscript. The main corrections in the paper and the responds to the reviewer’s comments are as following:
Dear Authors,
thank you very much indeed for your corrections introduced to the text.
I still have two minor remarks:
Question 1:
Please make Figures more clear.
Response:
Thank you very much for the good suggestion. We did everything we could to improve the Figures resolution, and the correction Figures were showed in the manuscript.
Additionally, we tried our best to improve the manuscript and made some changes in the manuscript. These changes will not influence the content of the paper. And here we did not list the changes but marked up with the "Track Changes" function in the revised manuscript.
We appreciate for Editors and Reviewers’ warm work earnestly, and hope that the correction will meet with approval.
Once again, thank you very much for your comments and suggestions.
Sincerely yours,
Lei Yang
Reviewer 2 Report
In the present manuscript, the extraction behaviors of 3 flavonoids, namely taxifolin, diosmin and quercetin, have been investigated in the leaves and bark of Abies nephrolepis. For this purpose, the following etraction parameters were evaluated: ethanol volume fraction, liquid-solid ratio, temperature, ultrasound irradiation power and time, ultrasound frequency.
This type of studies are suitable for RSM analysis. For example, the authors used varied levels for each of the tested parameters while the rest are kept constant. However, when testing the extraction kinetics the ethanol volume fraction is kept at 50% which is the optimum condition according to the results presented in 2.1.1, while liquid -solid ratio is kept at 20 mg/L (see line 130) which is not the optimum condition according to the results presented in 2.1.2.
Author Response
Dear Reviewer,
Thank you for your comments concerning our manuscript entitled “Ultrasound-assisted extraction of taxifolin, diosmin and quercetin from Abies nephrolepis (Trautv.) Maxim.: kinetic and thermodynamic characteristics” (ID: molecules-745420). Those comments are all valuable and very helpful for revising and improving our paper, as well as the important guiding significance to our researches. We have studied comments carefully and have made correction which we hope meet with approval. All revisions made to the manuscript have been marked up with the "Track Changes" function in the revised manuscript. The main corrections in the paper and the responds to the reviewer’s comments are as following:
Dear Authors,
thank you very much indeed for your corrections introduced to the text.
I have remarks below:
Question 1:
This type of studies are suitable for RSM analysis. For example, the authors used varied levels for each of the tested parameters while the rest are kept constant.
Response:
Thank you very much for the good suggestion. We have added sentences as follows: Section 2.4 Optimization of extraction procedure in line 327-390, and Section 3.7 Optimization of extraction procedure with RSM in line 472-486. Additionally, different from the single factor analysis, the maximum total yield of the 3 flavonoids under the optimal condition analyzed by RSM were revealed in Abstract as follows: “(According to the RSM analysis, under the optimal operation conditions (ethanol concentration of 50%, liquid solid ratio of 20 mL/g, frequency of 45 kHz, extraction time of 39.25 min, ultrasound irradiation power of 160 W and temperature of 332.19 K), the total yield of the 3 flavonoids were 100.93 mg/g ± 4.01 mg/g from the leaves of A. nephrolepis (with 31.03 ± 1.51 mg/g, 0.31 ± 0.01 mg/g, 69.59 ± 2.57 mg/g for taxfolin, diosmin and quercetin, respectively), and under the optimal operation conditions (ethanol concentration of 50%, liquid solid ratio of 20 mL/g, frequency of 45 kHz, extraction time of 36.80 min, ultrasound irradiation power of 150 W and temperature of 328.78 K), 16.05 mg/g ± 0.38 mg/g were obtained from the bark of A. nephrolepis (with 1.44 ± 0.05 mg/g, 0.47 ± 0.01 mg/g, 14.14 ± 0.38 mg/g for taxfolin, diosmin and quercetin, respectively), which were close to the prediction values.)” in line 24-33
Question 2:
However, when testing the extraction kinetics the ethanol volume fraction is kept at 50% which is the optimum condition according to the results presented in 2.1.1, while liquid -solid ratio is kept at 20 mg/L (see line 130) which is not the optimum condition according to the results presented in 2.1.2.
Response:
Thank you very much for the good suggestion. We did the ethanol concentration and liquid solid ratio optimization at Section 2.1.1 and Section 2.1.2, respectively. Firstly, ethanol concentration was the former section which was selected the optimal ethanol concentration of 50%, secondly, we discussed the optimal liquid solid ratio was 20 mL/g at Section 2.1.2.
We tried our best to improve the manuscript and made some changes in the manuscript. These changes will not influence the content of the paper. And here we did not list the changes but marked up with the "Track Changes" function in the revised manuscript.
We appreciate for Editors and Reviewers’ warm work earnestly, and hope that the correction will meet with approval.
Once again, thank you very much for your comments and suggestions.
Sincerely yours,
Lei Yang
Reviewer 3 Report
The purpose of manuscript ID 745420 was to select the parameters for the extraction of the leaves and bark of Abies nephrolepis giving optimal yield of the three flavonoids, including taxyfolin, diosmin and quercetin. However, some major comments and considerations should be taken before publishing in Molecules:
In the Introduction, the Authors provided the information on the chemical composition of A. nephrolepis but in my opinion this issue is too briefly presented. I suggested, that more information and references about types of polyphenols of subjected A. nephrolepis should be included in the Introduction. Please prove, based on literature data or preliminary HPLC analysis, that the flavonoids studied, including taxifolin, quercetin and diosmin are the dominant components of the flavonoid fraction of the A. nephrolepis leaves and bark. This is necessary to prove that the activity of these compounds mentioned in the Introduction might significantly affect the activity and use in medicine of the titled leaves and bark.
Please provide the sugar structure of diosmin (Figure 1) in the conformation of the chair. In the sugar structure presented by the Authors it is not clear whether the identified compound is glucose or, for example, galactose. It is well known that diosmin is diosmetin 7-O-rutinoside. Therefore, please replace 5,7,3-trihydroxy-4-methoxyflavone (line 53) by the common name
of this compound namely diosmetin.
To make the present study more interesting from the phytochemical point of view, the Authors should also add the figure showing the HPLC chromatograms of the polyphenol separation, that were observed under optimized extraction conditions.
In my opinion, in order to select the optimal extraction conditions for the most effective isolation of polyphenols, the Authors should employ the Response Surface Methodology (RSM) method. Currently, this method is commonly use and present in numerous articles published in Molecules (references below). I kindly ask you to comply with this request.
Response Surface Methodology Optimization of Ultrasonic-Assisted Extraction of Acer Truncatum Leaves for Maximal Phenolic Yield and Antioxidant Activity, Molecules 2017, 22(2), 232; https://doi.org/10.3390/molecules22020232;
Simultaneous Optimization of Ultrasound-Assisted Extraction for Flavonoids and Antioxidant Activity of Angelica keiskei Using Response Surface Methodology (RSM), Molecules 2019, 24(19), 3461; https://doi.org/10.3390/molecules24193461
Optimization and Formulation of Fucoxanthin-Loaded Microsphere (F-LM) Using Response Surface Methodology (RSM) and Analysis of Its Fucoxanthin Release Profile, Molecules 2019, 24(5), 947; https://doi.org/10.3390/molecules24050947
Author Response
Dear Reviewer,
Thank you for your comments concerning our manuscript entitled “Ultrasound-assisted extraction of taxifolin, diosmin and quercetin from Abies nephrolepis (Trautv.) Maxim.: kinetic and thermodynamic characteristics” (ID: molecules-745420). Those comments are all valuable and very helpful for revising and improving our paper, as well as the important guiding significance to our researches. We have studied comments carefully and have made correction which we hope meet with approval. All revisions made to the manuscript have been marked up with the "Track Changes" function in the revised manuscript. The main corrections in the paper and the responds to the reviewer’s comments are as following:
Dear Authors,
thank you very much indeed for your corrections introduced to the text.
I have some remarks below:
Question 1:
In the Introduction, the Authors provided the information on the chemical composition of A. nephrolepis but in my opinion this issue is too briefly presented. I suggested, that more information and references about types of polyphenols of subjected A. nephrolepis should be included in the Introduction. Please prove, based on literature data or preliminary HPLC analysis, that the flavonoids studied, including taxifolin, quercetin and diosmin are the dominant components of the flavonoid fraction of the A. nephrolepis leaves and bark.
Response:
Thank you very much for the good suggestion. We rewrite the relevant content of Introduction as follows:“ (Previous studies have showed that nearly thirty flavonoids are found in A. nephrolepis leaves and bark, such as taxifolin, diosmin and quercetin [6]. And the taxifolin, diosmin and quercetin showed broader commercial exploitation among the nearly thirty flavonoids, which gained widespread attention in medical community. Additionly, we added Figure 8 showing the chromatography analysis of mixture standard, leaves and barks extracts of A. nephrolepis.) ” in line 53-56
Question 2:
This is necessary to prove that the activity of these compounds mentioned in the Introduction might significantly affect the activity and use in medicine of the titled leaves and bark.
Response:
Thank you very much for the good suggestion. We have added a sentence as follows: “(Reports revealed that the phenolics from leaves of A. nephrolepis showed excellent antioxidant in vitro [21]and anti-hepatoma in vivo[22].)” in line 68-69.
Question 3:
Please provide the sugar structure of diosmin (Figure 1) in the conformation of the chair. In the sugar structure presented by the Authors it is not clear whether the identified compound is glucose or, for example, galactose. It is well known that diosmin is diosmetin 7-O-rutinoside. Therefore, please replace 5,7,3-trihydroxy-4-methoxyflavone (line 53) by the common name
of this compound namely diosmetin.
Response:
Thank you very much for the good suggestion. We have corrected the Figure 1 in line 63-64 and the sentence as follows: “(Diosmin, the diosmetin 7-O-rhamnosylglucoside (Figure 1), has been proposed as a drug due to its antitumor effects.)” in line 61-62.
Question 4:
To make the present study more interesting from the phytochemical point of view, the Authors should also add the figure showing the HPLC chromatograms of the polyphenol separation, that were observed under optimized extraction conditions.
Response:
Thank you very much for the good suggestion. We have added Figure 8 to show the HPLC chromatograms of the polyphenol separation in line 460-461.
Question 5:
In my opinion, in order to select the optimal extraction conditions for the most effective isolation of polyphenols, the Authors should employ the Response Surface Methodology (RSM) method. Currently, this method is commonly use and present in numerous articles published in Molecules (references below). I kindly ask you to comply with this request.
Response Surface Methodology Optimization of Ultrasonic-Assisted Extraction of Acer Truncatum Leaves for Maximal Phenolic Yield and Antioxidant Activity, Molecules 2017, 22(2), 232; https://doi.org/10.3390/molecules22020232;
Simultaneous Optimization of Ultrasound-Assisted Extraction for Flavonoids and Antioxidant Activity of Angelica keiskei Using Response Surface Methodology (RSM), Molecules 2019, 24(19), 3461; https://doi.org/10.3390/molecules24193461
Optimization and Formulation of Fucoxanthin-Loaded Microsphere (F-LM) Using Response Surface Methodology (RSM) and Analysis of Its Fucoxanthin Release Profile, Molecules 2019, 24(5), 947; https://doi.org/10.3390/molecules24050947
Response:
Thank you very much for the good suggestion. We have added sentences as follows: Section 2.4 Optimization of extraction procedure in line 327-390, and Section 3.7 Optimization of extraction procedure with RSM in line 472-486. Additionally, different from the single factor analysis, the maximum total yield of the 3 flavonoids under the optimal condition analyzed by RSM were revealed in Abstract as follows: “(According to the RSM analysis, under the optimal operation conditions (ethanol concentration of 50%, liquid solid ratio of 20 mL/g, frequency of 45 kHz, extraction time of 39.25 min, ultrasound irradiation power of 160 W and temperature of 332.19 K), the total yield of the 3 flavonoids were 100.93 mg/g ± 4.01 mg/g from the leaves of A. nephrolepis (with 31.03 ± 1.51 mg/g, 0.31 ± 0.01 mg/g, 69.59 ± 2.57 mg/g for taxfolin, diosmin and quercetin, respectively), and under the optimal operation conditions (ethanol concentration of 50%, liquid solid ratio of 20 mL/g, frequency of 45 kHz, extraction time of 36.80 min, ultrasound irradiation power of 150 W and temperature of 328.78 K), 16.05 mg/g ± 0.38 mg/g were obtained from the bark of A. nephrolepis (with 1.44 ± 0.05 mg/g, 0.47 ± 0.01 mg/g, 14.14 ± 0.38 mg/g for taxfolin, diosmin and quercetin, respectively), which were close to the prediction values.)” in line 24-33
We tried our best to improve the manuscript and made some changes in the manuscript. These changes will not influence the content of the paper. And here we did not list the changes but marked up with the "Track Changes" function in the revised manuscript.
We appreciate for Editors and Reviewers’ warm work earnestly, and hope that the correction will meet with approval.
Once again, thank you very much for your comments and suggestions.
Sincerely yours,
Lei Yang
Round 2
Reviewer 2 Report
The authors have carried out the suggested corrections. Therefore, I recommend the aceptance of the manuscript in its current form.
Author Response
Thanks a lot.
Reviewer 3 Report
I recommend this manuscript to be published in the Molecules, but after correcting one more error. In the Figure 1, the Authors present the formula of quercitrin (quercetin 3-rhamnoside). In the previous version ofmanuscript, the formula of flavonol-type aglycon quercetin, that is also in the text, was placed here. Please correct the flavonoid formula in the Figure 1 or correct the name of the flavonoid compound in the submitted manuscript.
Author Response
Dear Reviewer,
Thank you for your comments concerning our manuscript entitled “Ultrasound-assisted extraction of taxifolin, diosmin and quercetin from Abies nephrolepis (Trautv.) Maxim.: kinetic and thermodynamic characteristics” (ID: molecules-745420). Those comments are all valuable and very helpful for revising and improving our paper, as well as the important guiding significance to our researches. We have studied comments carefully and have made correction which we hope meet with approval. All revisions made to the manuscript have been marked up with the "Track Changes" function in the revised manuscript. The main corrections in the paper and the responds to the reviewer’s comments are as following:
Dear Authors,
thank you very much indeed for your corrections introduced to the text.
I have 1 minor remark:
Question 1:
I recommend this manuscript to be published in the Molecules, but after correcting one more error. In the Figure 1, the Authors present the formula of quercitrin (quercetin 3-rhamnoside). In the previous version of manuscript, the formula of flavonol-type aglycon quercetin, that is also in the text, was placed here. Please correct the flavonoid formula in the Figure 1 or correct the name of the flavonoid compound in the submitted manuscript.
Response:
We are so sorry that we made the mistake, and we have made correction of this Figure and marked up with the "Track Changes" function in the revised manuscript.
We appreciate for Editors and Reviewers’ warm work earnestly, and hope that the correction will meet with approval.
Once again, thank you very much for your comments and suggestions.
Sincerely yours,
Lei Yang